# COACH: Cooperative Robot Teaching

**Cunjun Yu**[1]    **Yiqing Xu**[1]    **Linfeng Li**[1]    **David Hsu**[1,2]

[1]School of Computing
[2]Smart Systems Insitute
National University of Singapore

**Abstract:** Knowledge and skills can transfer from human teachers to human students. However, such direct transfer is often not scalable for *physical* tasks, as they require one-to-one interaction, and human teachers are not available in sufficient numbers. Machine learning enables robots to become experts and play the role of teachers to help in this situation. In this work, we formalize *cooperative robot teaching* as a Markov game, consisting of four key elements: the target task, the student model, the teacher model, and the interactive teaching-learning process. Under a moderate assumption, the Markov game reduces to a partially observable Markov decision process, with an efficient approximate solution. We illustrate our approach on two cooperative tasks, one in a simulated video game and one with a real robot.

**Keywords:** Robot Teaching, Human-Robot Interaction

## 1  Introduction

How do we teach humans to re-orientate a table jointly or play tennis? Humans often learn by practicing the skills with teachers or partners [1, 2, 3]. This mode of learning is, however, difficult to scale up, as it requires one-to-one interaction and there are not sufficient human teachers [4]. With advances in machine learning, robots can not only master complex tasks [5, 6, 7] but also collaborate with humans and adapt to human behaviors [8, 9, 10]. In this work, we aim to create *robot teachers* for physical tasks, thus scaling up teaching and providing learning opportunities to many even when human teachers are not available.

Specifically, we propose *Cooperative rObot teACHing* (COACH), a robot teaching framework to teach humans cooperative skills for two-player physical tasks through interaction. We assume the robot teacher has full knowledge of the task, specifically, a set of optimal policies. The objective is to teach the student an optimal policy as fast as possible. See Fig. 1 for an illustration. COACH treats the teaching task as a two-player Markov game for a *target task*. One player is the robot teacher, and the other is the human student. Under a suitable student learning model, COACH transforms the game into a *partially observable Markov decision process* (POMDP). The POMDP solution enables the robot teacher to adapt to the different behaviors, according to the history of interactions.

One key challenge of COACH is to represent the student's knowledge of the target skills and learning behaviors. First, we leverage *item response theory* (IRT), a well-established framework for educational assessment [11]. IRT provides simplified parametric models that capture the student's knowledge level with respect to the task difficulty in a small number of parameters. COACH treats these parameters as latent variables in the teaching POMDP and learns them from human-robot interaction data by solving the POMDP. Next, to teach complex skills, we draw insights from student-centered learning [12] and human-robot cross-training [13]. We decompose a complex target skill into a set of sub-skills, based on the student's potential roles in the target task. With this compact, decomposed skill representation, we naturally obtain a *partially assistive* robot teaching curriculum to facilitate learning: the human student learns the sub-skills one at a time, and the robot teacher assists with the sub-skills not yet learned, to complete the target task. While the robot assists the human in the teaching task, its behavior differs from those in common collaborative human-robot interaction tasks [14, 15]. There, the primary objective is to complete the task, and the robot is fully assistive: if the human does not perform, the robot then tries to complete the task on its own, if

6th Conference on Robot Learning (CoRL 2022), Auckland, New Zealand.

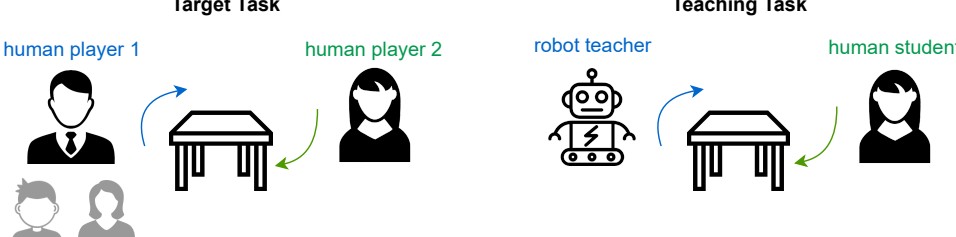

Figure 1. Cooperative robot teaching. In the target task (left), two human players jointly reorient a table, for example. In the corresponding teaching task (right), the robot teacher interacts with the human student and teaches cooperative skills so that the student learns to cooperate with partners with varying capabilities or preferences in the target task.

possible. In the teaching task, the robot is partially assistive and usually avoids assisting with the specific sub-skill to be learned, in order to encourage student exploration and learning.

As a first attempt, we conducted human-subject experiments on two challenging human-robot collaboration tasks, Overcooked-AI and Cooperative Ball Maze ( Fig. 2). Our results show that COACH enables the robot teacher to model and reason over adaptive human students in cooperative teaching. Also, a fully-assistive teacher may impede student learning, and a partially assistive teacher indeed motivates the student to explore new strategies.

## 2   Related Work

**Assistance in HRI.** One major aspect of HRI is how the robot could assist humans with a hidden human objective [14, 15]. The objective of the robot is to infer the human's intention and learns to assist the human. In its simplest form, the action selection and human intention inference are separated [16, 17, 18]. A decision-theoretic framework, assistant POMDP, is developed to capture the general notion of assistance in HRI [19]. The robot integrates the reward learning and control modules to perform sophisticated reasoning over human feedback [20, 21]. However, these two approaches neglect human learning/adaptation and may hinder humans from improving their skills. Our work focuses on how to generate behaviors that facilitate human learning during interactions.

**Collaboration in HRI.** Another important aspect of HRI is to model interactions as the collaboration between the human and the robot [22], for which the human and the robot share the same objective. However, the joint optimal policy, e.g. rotating the table counter-clockwise, is unknown to both agents in the first place. Their interaction is mutually adaptive [23, 24, 25]. Particularly, as pointed out in [10], if one side is only aware of partial information about the task, the optimal policy pair naturally induces the behavior of active teaching, active learning, and efficient communication between the robot and human. In this work, we focus on the following setting: given that the robot teacher knows the optimal policy, how to carry out active teaching.

**Teaching Algorithm for Algorithms.** Teaching for algorithms aims to facilitate the learning of the algorithm by choosing or generating training samples. Various teaching techniques including curriculum learning [26] and machine teaching [27, 28, 29, 30, 31] have been effectively applied to supervised learning and semi-supervised learning problems. Similar ideas are further extended to train reinforcement learning agents to learn complex skills, e.g., generate training environment for reinforcement learning [32, 33, 34], choose various demonstrations [35] or learn to decompose the skill [36, 37]. Teaching in cooperative multi-agent RL allows agents to simultaneously become teachers and students for each other [3, 38, 39]. However, such approaches generally require relatively more data for training and to some extent the controlled learning behavior of the learner. Transfer of these approaches to human learning is promising but difficult.

**Teaching Algorithm for Human.** Despite the aforementioned practical challenges, some algorithms have been successfully deployed for human learning. Attempts on teaching the crowd on classification or concepts prove to be successful [40, 41, 42, 43, 44]. While humans can learn concepts from visual or verbal examples, complex skills like motor control skills can hardly be mastered through these signals. Recently, skill discovery techniques in reinforcement learning have been in-

troduced to generate a curriculum based on skill decomposition and facilitate humans to learn motor control skills [45]. It focuses on how to adaptively decompose the skill into learnable sub-skills for a human to practice on its own and achieves promising results. Here, we seek to automate the teaching process for humans to cooperate in a physical task and provide a framework for this teaching mode, e.g., table co-reorientation.

## 3 Cooperative Robot Teaching

We identify four key elements in COACH: (1) the target task, (2) the student learning model, (3) the teacher model, and (4) the interactive teaching-learning process.

**Target task.** In this work, we focus on teaching in *a two-player cooperative task*, which we call it the *target task*.

**Definition 1** . *The target task is a two-player cooperative Markov game $\mathcal{M} = (S, A^1, A^2, T, R, \gamma)$ between two agents,* 1 *and* 2*, where*

- *$S$ is a set of target task states;*
- *$A^1$ is a set of actions for agent* 1*;*
- *$A^2$ is a set of actions for agent* 2*;*
- *$T(s'|s, a^1, a^2)$ is a conditional probability function on the next target task state $s' \in S$, given the current state $s \in S$ and both agents' actions $a^1 \in A^1$ and $a^2 \in A^2$;*
- *$R(s, a^1, a^2)$ is a target task reward function that maps the target task state and players' actions to a real number;*
- *$\gamma$ is a discount factor.*

At each step $t$, agent 1 and 2 both observe the current task state $s_t$ and select their respective actions $a_t^1 \sim \pi^1$ and $a_t^2 \sim \pi^2$, where $\pi^i$ the policy of agent $i$ for $i = 1, 2$. They then receive a joint reward $r_t = R(s_t, a_t^1, a_t^2)$. The next state is updated as $s_{t+1} \sim T(s_{t+1} \mid s_t, a_t^1, a_t^2)$.

Given the definition of the target task, we first answer how to represent the knowledge/skills. In this work, we choose to represent a *skill* by the optimal policy $\phi^*$ to the target task. The optimal policy maximizes the expected cumulative reward when the agent is cooperating with a given partner. For example, in the table co-reorientation task, the agent needs to learn to deal with either stubborn or adaptive partners. We recognize that there are other ways to represent knowledge/skills, such as a set of demonstrations and the ground-truth reward function. However, such representations are indirectly linked with the skill's performance; therefore, evaluating its proficiency is more obscured. We choose the optimal policy as the representation since it can be directly optimized over and evaluated.

**Student.** The student policy is non-stationary since it will improve along with teaching. We model this evolutionary behavior with a tuple of student policy and an updating function $(\phi, U)$. The student policy represents the student knowledge state. It will take in the current target task state $s$ as input and output the student's action. The updating function $U$ models how the student changes its policy after each teaching step.

**Teacher.** We define the teacher as a knowledgeable agent (expert) who knows a set of optimal policies $\Phi^*$ for a target task. The teacher aims to acquire a teaching policy $\pi^{\mathrm{T}}$ that can teach any $\phi_i^* \in \Phi^*$ to the student effectively. In this general setting, the choice of the optimal student policy $\phi^*$ depends on the capability, preference, and current knowledge level of the student. A principled approach to selecting the optimal student policy needs to consider the student's preference, his/her update model for the knowledge level, and an estimate of his/her current capability. In this paper, we assume that we have an oracle to choose the optimal student policy $\phi^* \in \Phi^*$ to teach, such that this policy $\phi^*$ matches the preference of the student. The teacher can be described by a tuple of an optimal target task policy and the corresponding teaching policy, $(\phi^*, \pi^{\mathrm{T}})$.

**Interactive teaching-learning**. We now refer to the robot as the teacher and the human as the student. In the target task, the teacher knows the target task's optimal policy $\phi^*$ while the student does not. The teacher's goal is to act in the most informative way so that the student learns $\phi^*$ fastest. The choice of $\phi^*$ should account for the student's preferences. To embed the objective of teaching and distinguish it from the *Target Task*, we define it as the *Teaching Task* in the following way:

**Definition 2** . *Given a target task $\mathcal{M} = (S, A^1, A^2, T, R, \gamma)$, a student $(\phi, U)$, and an optimal policy $\phi^*$ for the target task, the teaching task is a POMDP $\mathcal{M}' = (\bar{S}, \bar{A}, \bar{T}, \bar{O}, \bar{Z}, \bar{R}, \bar{\gamma})$ for the teacher, where*

- *$\bar{S}$ is a set of teaching states: $\bar{s} = (s, \phi)$, for target task state $s \in S$ and student policy $\phi$;*
- *$\bar{A}$ is a set of actions: $\bar{A} = A^1 \cup A^2$;*
- *$\bar{T}(\bar{s}' \mid \bar{s}, \bar{a})$ is a conditional probability function on the next state $\bar{s}' \in \bar{S}$, given the current state $\bar{s} \in \bar{S}$ and teacher's action $\bar{a} \in \bar{A}$;*
- *$\bar{O}$ is a set of observations: $\bar{o} = (s, r)$, for target task state $s \in S$ and target task reward $r$;*
- *$\bar{Z}(\bar{o} \mid \bar{a}, \bar{s})$ is a conditional probability function on the observation $\bar{o} \in \bar{O}$, given teacher's action $\bar{a} \in \bar{A}$ and current state $\bar{s} \in \bar{S}$;*
- *$\bar{R}(\bar{s}, \bar{a}, \bar{s}')$ is a teaching reward function that maps current state $\bar{s} \in \bar{S}$, teacher's action $\bar{a} \in \bar{A}$, and next state $\bar{s}' \in \bar{S}$ to a real number measuring the effectiveness of teaching;*
- *$\bar{\gamma}$ is a discount factor.*

The objective of the teaching task is to derive a teaching policy $\pi^{\mathrm{T}}$, enabling students to learn $\phi^*$ for the target task fastest. More specifically, the teacher can influence the student through interactive actions $\bar{a} \in \bar{A}$. Given the student policy $\phi$ and the update function $U$, the student would learn through interaction: $\phi_{t+1} = U(\phi_t, \cdot)$. The goal of the teaching task is to find a teaching policy $\pi^{\mathrm{T}}$ that allows $\phi_0 \to \phi^*$ as fast as possible.

Next, we introduce our choice of the teaching policy $\pi^{\mathrm{T}}$, the update function $U$, and the reward function $\bar{R}$. To devise a student-aware teaching strategy, apart from the current state $s_t$ and the target policy $\phi^*$, our $\pi^{\mathrm{T}}$ also takes the history of observation as input. The history of observation is $h_t = [(s_0, r_0), ..., (s_t, r_t)]$. The action of the teacher can be sampled from the policy, i.e., $\bar{a}_t \sim \pi^{\mathrm{T}}(\bar{a}_t \mid h_{t-1}, s_t, \phi^*)$. The student updates $\phi$ with any arbitrary iterative functions conditioned on the history of interactions: $\phi_{t+1} = U(\phi_t, h_t)$. Moreover, to incentivize the teacher to speed up the teaching process, we introduce a step-wise teaching cost to the teacher $c_t = C(s_t, \bar{a}_t)$ to penalize unnecessary teaching actions. To this end, we define the reward function as

$$\bar{R}(\bar{s}, \bar{a}_t, \bar{s}'; D, C, \phi^*, \omega) = D(\phi_t, \phi^*) - D(\phi_{t+1}, \phi^*) - \omega C(s_t, \bar{a}_t), \tag{1}$$

where $\bar{s} = (s_t, \phi_t)$, $\bar{s}' = (s_{t+1}, \phi_{t+1})$, $\omega$ is the weighting factor to trade-off the teaching cost and teaching efficiency, and $D$ can be any reasonable distance measure between two policies, e.g., initial state value in the target task. The solution to the POMDP $\mathcal{M}'$ is a teaching policy $\pi^{\mathrm{T}}$ that maximizes the expected sum of rewards $\mathbb{E}_{\bar{a}_t \sim \pi^{\mathrm{T}}}[\sum_{t=0}^{\infty} \bar{\gamma}^t \bar{R}(\bar{s}, \bar{a}_t, \bar{s}')]$.

## 4 Method

In this section, we provide a solution that grounds all the ingredients in the conceptual framework of COACH. The main spirit of our solution is to parameterize students' knowledge state with IRT and decompose complex tasks into a set of role-based independent skills. This enables easier evaluation of the students' proficiencies and provides a ground to derive the partially assistive interaction mode. To begin with, we first define the action space, $\bar{A}$.

### 4.1 Actions

The action space is constructed through sub-skill decomposition. Sub-skills decomposition is well-studied for single-

---

**Algorithm 1** Approximated Solution to the Teaching Task

**Require:** Maximum Interactions $L$, Predefined Interactions $N$
1: **for** $k \in \bar{A}$ **do**:
2:     Randomly initialize $\lambda$ and $\alpha_t$, $\beta$, and $\mathbb{X} = \{\}$
3:     **for** $i = 1, 2, ..., N$ **do**:
4:         $\mathbb{X}.\mathtt{add}(v_i)$
5:     **end for**
6: **end for**
7: **for** $i = 1, 2, ..., L$ **do**:
8:     **for** $k \in \bar{A}$ **do**:
9:         Learn $\lambda$ and $\alpha_t$, $\beta$ from $\mathbb{X}$
10:     **end for**
11:     $k \leftarrow$ Action selection from $\lambda$ and $\alpha_t$, $\beta$
12:     $v_i \leftarrow$ Performance measure from interactions
13:     $\mathbb{X}.\mathtt{add}(v_i)$
14: **end for**

---

agent tasks [46, 47, 48]. However, extending the same idea to a multi-agent setting is still challenging since task completion relies on the interaction among multiple parties. We observe that in a multi-agent game, the task naturally comprises several roles, of which each agent takes a subset. The well-established leader-follower model is a particular choice of role-based skill decomposition [49, 50, 51, 52]. Therefore in our work, we propose to decompose skills based on role alloca-

tion. We divide the skill into $K$ independent teachable sub-skills according to the student's potential roles in the task. The teacher's action space $\bar{A} = \{k : k \in \mathbb{Z}, 0 \le k < K\}$ consists of teaching each sub-skill. Such a decomposition of skills naturally leads to a partially assistive mode of interaction.

## 4.2 States

The state space is constructed with Item Response Theory (IRT) [11]. IRT provides a parametric form to represent students' skill levels. Given the limited interactions, we adopted the simplest form, the one-parameter logistic model (1PL), to model human skills. In the 1PL model, each sub-skill $k \in \bar{A}$ is assigned a parameter $\beta^k$ representing the difficulty, and a parameter $\alpha^k$ called the *proficiency* representing a student's knowledge state. The probability that a student has mastered sub-skill $k$ is given by $P(k) := \sigma(\alpha^k - \beta^k)$, where $\sigma$ is the sigmoid function. Hence, instead of representing the state with the student's policy $\phi$, we use $(\alpha, \beta)^K$ to represent the hidden state. That is, for $\bar{s} \in \bar{S}, \bar{s} = (s, (\alpha, \beta)^K)$, where $(\alpha, \beta)^K$ is hidden. For each student and each $k \in \bar{A}$, we assume that $\alpha$ changes over time while $\beta$ does not.

## 4.3 Transitions

The transition model consists of two main parts, the target task transition model $T$, and the student's update function $U$. While the former one is known to the teacher, we need to make assumptions about the latter one. Since we define the state space over the student's proficiency $\alpha$ in Sec 4.2, the transition model is also constructed over the proficiency. Following the previous work on online estimation of student proficiency [53, 54], for each sub-skill, we model the student's proficiencies over time as a Wiener process: $U(\alpha_{t+\Delta t}|\alpha_t) = \exp\left(-\frac{(\alpha_{t+\Delta t} - \alpha_t)^2}{2\lambda \Delta t}\right)$, where $\Delta t$ refers to the step interval and $\lambda$ is a parameter controlling the "smoothness" with which student's proficiency varies over time. For each student and for each $k \in \bar{A}$, we assume $\lambda$ does not change over time and is learned for each sub-skill respectively. To this end, we construct the transition model in the POMDP as $\bar{T} = \{T, U\}$, where $T$ is the transition function in the target task.

## 4.4 Observations

The observation is composed of the target task state and the reward received, $(s, r)$. Recall that in Sec 4.1, we define the action as choosing one sub-skill to train the student, which is a macro-action. For teaching sub-skill $k$, we redefine the observation as the ratio between the target task rewards achieved by the student's current and optimal policies: $v := \frac{R(s, \bar{a}, a^S)}{R(s, \bar{a}, a^*)}$, where $a^*$ is the action generated by the optimal policy $\phi^*$ given the same the target task state $s$. Since all the sub-skills are treated equally, we will omit the index $k$ for simplicity in the following discussion. As a result, for $\bar{o} \in \bar{O}, \bar{o} = (s, v)$. Unlike the binary response in conventional knowledge tracing, the response $v$ we have is continuous and we assume the teacher will only teach one sub-skill at a time. Thus, we use the continuous Bernoulli distribution to construct the observation model: $Z(v|P(k)) = P(k)^v(1 - P(k))^{1-v}$, where $k$ is the sub-skill being taught when $v$ is observed. As a result, the observation model can be defined as $\bar{Z} = \{I, Z\}$, where $I$ is an identity mapping for the observable target task state, $I(s) = s$.

## 4.5 Reward

The distance between the student's policy and the optimal policy can be represented using $P(k)$. We represent the distance as the average of one minus master probabilities of each sub-skill: $D(\phi, \phi^*) = \frac{\sum_{k=0}^{K} 1 - P(k)}{K}$. There are other ways to specify the goal according to the decomposition of the skill, e.g. weakest or multiply [55]. We choose the sum due to our independence assumption on sub-skills. In this work, we assume the cost is uniform, thus, given a finite horizon of interactions, maximizing the reward function defined in Equation (1) is equivalent to maximizing $\bar{R}(\bar{s}, \bar{a}_t, \bar{s}') = \frac{\sum_{k=0}^{K} P_{t+1}(k) - P_t(k)}{K}$, where $P_t(k) = \sigma(\alpha_t^k - \beta^k)$.

## 4.6 Model Learning and Decision Making

We use the student's performance during the interactions to estimate both $\lambda$ and $\alpha_t, \beta$. Parameters for each sub-skill are learned separately, thus, we omit $k$ for simplicity. Let $v_{1:t}$ denote sequences of student's performance measure against the optimal policy up to step $t$. We have the posterior $P(\lambda, \alpha_t, \beta|v_{1:t}) \propto P(v_{1:t}|\lambda, \alpha_t, \beta)P(\lambda, \alpha_t, \beta)$. The conditional probability of the observation and current proficiency can be obtained by integrating out all the previous proficiencies. The likelihood can be approximated through $P(v_{1:t}|\lambda, \alpha_t, \beta) \approx \prod_{t'=1}^{t} \int P(v_{t'}| \lambda, \alpha_{t'}, \beta)U(\alpha_{t'}|\alpha_t)\mathrm{d}\alpha_{t'}$. An approximation of the log posterior over the student's current proficiency given previous responses can be derived to learn the parameters $\lambda$ and $\alpha_t, \beta$. Following [53, 54], we employ maximum a posteriori estimation (MAP) to learn these parameters. Given the estimation of current state using the past history, we use one-step look-ahead to reduce the impact of the inaccuracy in the transition function. At timestep $t$, the teacher's action is given as

$$\bar{a}_{t+1} = \arg\max_{k \in \bar{A}} \int U(\alpha_{t+1}^k|\alpha_t^k)P_{t+1}(k) \, \mathrm{d}\alpha_{t+1}^k - P_t(k). \tag{2}$$

In practice, the student is asked to perform on each sub-skill for a few interactions to initialize the parameters.

## 4.7 Training on Sub-skills

Our overall strategy for training students on each sub-skill is to diversify scenarios the student would encounter during training. Training students on sub-skills naturally leads to a partially assistive partner on unlearned sub-skills, which allows the student to explore the sub-skill freely. We adopt an intuitive assumption: *an agent learns cooperation better with a diverse group of partners*. Such a teaching strategy is effective when dealing with synthetic students [56, 57]. The student could learn from a diverse set of partially assistive partners or learn to cope with them by acquiring new skills.

# 5 Experiments

We carried out two human-subject experiments to demonstrate how COACH works, one in simulation (Overcooked-AI [58]) and the other with a real robot (Cooperative Ball Maze). Experiment setups are shown in Figure 2. We investigated the teaching performances of three types of teachers: the **fully-assistive** teacher who performs optimally concerning the student's initial capability, the **student-aware** teacher who behaves according to our teaching strategy, and the **random** teacher. The random teacher in the Cooperative Ball Maze experiment chooses sub-skills randomly, while the random teacher in the Overcooked-AI experiment executes actions randomly.

## 5.1 Setups

**Overcooked-AI.** Overcooked-AI is a benchmark environment for fully cooperative human-AI task performance and has become a well-established domain for studying coordination [59, 60, 61, 62]. The goal of the game is to cook and deliver as much soup as possible in a limited time. We decompose the policy into two sub-skills: *putting ingredients in the pot* and *delivering the soup*. To put ingredients in the pot, there exists one *efficient strategy* to pass the ingredient through the middle table. In brief, rather than picking up one onion at a time and putting them into the pot, the efficient strategy is 1) put multiple onions on the middle table; 2) go to the pot; 3) pick up onions from the middle table; 4) put them into the pot. The overall idea is to reduce the number of movements needed to deliver the same

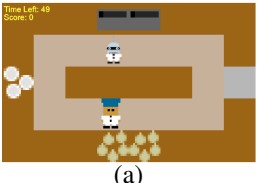 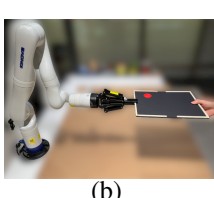

(a)        (b)

Figure 2. Experiment setups. (a) Overcooked-AI layout: human participants control the "chef" and the robot controls the "robot". (b) The real robot setup of Cooperative Ball Maze with a simplified setting.

amount of ingredients. We recruited $N=20$ (8 females and 12 males) participants and randomly assigned them into three groups, each with a different teaching strategy. Students are trained with different teachers and are evaluated with a common unseen partner. We emulate the human partner

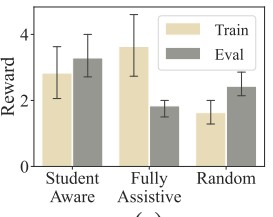 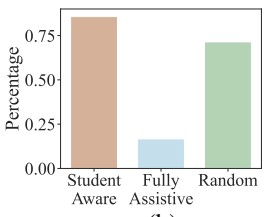 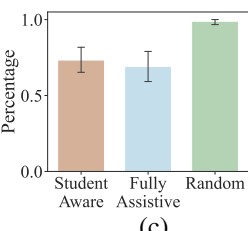

(a)          (b)          (c)

Figure 3. Results of the Overcooked-AI experiment. (a) Rewards achieved together by the human-robot pairs during training and evaluation. The error bars correspond to the 95% confidence intervals (95%CI). The student-aware teacher outperformed the fully assistive and the random teachers in terms of the evaluation reward (with one-sided $p$-values 0.001 and 0.03). (b) Percentage of students who found the efficient strategy. None of the students are aware of this strategy at the beginning of the training. (c) Percentage of reward achieved by the human participants during training.

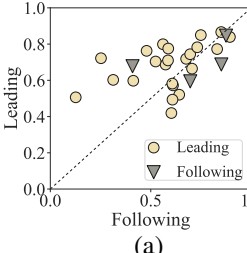 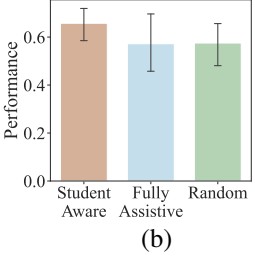 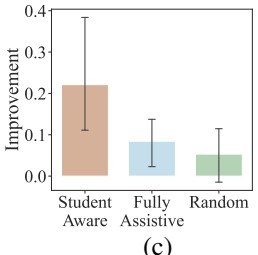

(a)          (b)          (c)

Figure 4. Results of the Cooperative Ball Maze experiment. (a) Evaluation of performances of the two sub-skills of all participants. The marker styles correspond to the sub-skill preferences of the participants. (b) Evaluation performances. The error bars correspond to the 95%CIs. (c) Improvements after 20 interactions. The error bars correspond to the 95%CIs. The students improve more under student-aware teachers than both fully-assistive and random teachers (with one-sided $p$-values 0.069 and 0.039).

in evaluation using a trained model. Each participant was trained for 5 games and then evaluated for 1 game.

**Cooperative Ball Maze.** The Cooperative Ball Maze game requires coordination from both the robot and the human. Each party will hold one side of the maze board and tilt it to move the ball out from one of the two exits. We define two sub-skills *leading the rotation* and *following the rotation*. We recruited $N$=21 (10 females and 11 males) participants to carry out human-subject experiments. The participants were first evaluated in the two sub-skills, then trained for 20 interactions, and finally evaluated in the two sub-skills again. Details can be found in the supplementary materials.

## 5.2 Results

*A fully-assistive teacher impedes human's acquisition of skills.* In the Overcooked-AI experiment shown in Figure 3(a), we observe that the students trained with a fully-assistive teacher perform worse than the students with a random teacher: it seems that a student becomes "lazy" and free rides the teacher when the teacher unilaterally adapts to the student and performs optimally. We further investigate the learning pattern of the "lazy student" problem and find out that *this "laziness" does not lie in the student's reluctance to take actions, but rather in the lack of motivation to explore and improve*. In Figure 3(c), we show the percentage of reward achieved by the student in Overcooked-AI during training. Compared with the student-aware counterpart, the percentage of reward achieved by humans is similar. However, only 17% of the participants of the group find out the efficient strategy (Figure 3(b)), which is crucial to achieving high scores in the evaluation.

*Partially assistive or random partner motivates students to explore new strategies.* By leaving some/all work to the student, partially assistive and random teachers both motivate the student to acquire new skills. This is shown in Figure 3(b) that most of the students under these two teachers can find out the efficient strategy in Overcooked-AI. However, their performance and the robustness of the learned strategies differ significantly. Though multiple explanations could account for it, we hypothesize the student under the random teacher learns a single fixed strategy to finish the task alone (Figure 3(c)). Such a strategy that completes the task alone cannot utilize the possibly helpful inputs from the partner, therefore resulting in a poorer performance score.

*An individualized curriculum should be designed for the student.* In the post-experiment survey of Cooperative Ball Maze, we asked the participants "which mode of the robot is easier to cooperate with?". Out of the 21 participants, 4 participants preferred to follow the robot and 17 participants preferred to lead the robot. Moreover, as we evaluated the student performance with partners of different sub-skills, we found that the student performances were consistent with their declared preferences (Figure 4(a)). That is to say, the student may have a bias over which strategy to acquire, and tailoring the teaching curriculum to focus on that specific strategy is efficient and more intuitive to the student. As demonstrated in Figure 5(a), after the first 6 trials that estimated the student's proficiency for each sub-skill, the teacher found out this student improved more as the leader, therefore, the teacher allocated 10 trials to perfect the *leading* sub-skills and only 4 trials for *following*. Moreover, one participant in the random teacher group responded "the robot leading mode is too difficult and I gave up". This demonstrates the importance of an individualized curriculum: though there are multiple equally optimal strategies, the individual may have strong preferences, and teaching a non-preferable strategy will discourage the student from learning anything at all. We refer the readers to the Appendix for the complete data of all participants.

## 6 Limitation

**Decomposition into sub-skills.** For many tasks, it is not easy to identify distinct roles to fulfill the local-independence criteria of sub-skills. We manually decompose the skill into a few sub-skills according to the role of the student. Often, such a decomposition may not be possible or requires careful design. We can mitigate this problem with recent progress on skill decomposition in single-agent task [45] and role-based task decomposition in multi-agent tasks [63]. However, it still demands much more effort to verify their efficacy with a real human on real-world tasks.

**Teacher's Knowledge.** In the definition of the teaching task, we assume the teacher has full knowledge of optimal policies. However, it can be hard for the robot to know the oracle human policy beforehand.

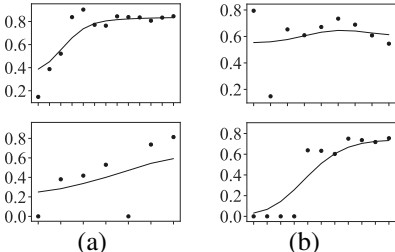

(a)          (b)

Figure 5. Sub-skill performances (vertical axis) with respect to training progress (horizontal axis) of two example participants trained by the student-aware teacher. Dots represent the raw scores and lines represent the smoothed scores. The top and bottom figures correspond to leading and following sub-skills respectively. (a) Participant 4. The student improved more when trained in the leading sub-skill. (b) Participant 6. The student improved more when trained in the following sub-skill.

To make the conceptual framework practical, we need to relax the requirement on the teacher's prior knowledge. In our implementation, we reduce such an assumption by approximating the distance to the optimal policy by the difference in performances. There can be cases where the optimal performance is hard to know or such relaxation results in severe information loss. We need more insights on tasks to make the framework practical.

**Curriculum design.** In this work, we only design the curriculum over different sub-skills. However, during our experiment, we observe that humans show various responses to the same sub-skill of different difficulties. One specific finding is that people may give up learning when the task becomes too difficult. As a result, a finer-grained curriculum on the sub-skill training shall be generated to further facilitate human learning.

## 7 Conclusion

In this work, we propose a conceptual framework, Cooperative Robot Teaching, that enables robots to teach humans in cooperative tasks. We show that, by abstracting a teaching task over the original duo cooperative task, the robot can learn to act as a specialized teacher to humans. To be more specific, we model the teaching task as a POMDP with hidden student policy and propose a partially assistive teaching curriculum to support human learning. We believe that robot teaching fills in the gap in the bilateral knowledge transfer in HRI: unlike other HRI tasks where the humans instruct the robots how to behave, the role is reversed and robots try to instill the knowledge back into humans. Despite the challenges that lie ahead, we believe that robot teaching has great potential and is a necessary step forward to bring robots closer to our daily life.

**Acknowledgements.** This research is supported in part by the National Research Foundation, Singapore under its Medium Sized Centre Program, Center for Advanced Robotics Technology Innovation (CARTIN), and AI Singapore Programme (AISG Award No: AISG2-PhD-2022-01-036[T] and AISG2-PhD-2021-08-014), and by the Science and Engineering Research Council, Agency of Science, Technology and Research, Singapore, under the National Robotics Program (Grant No. 192 25 00054).

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
