# OpenReview forum: "COACH: Cooperative Robot Teaching"
_robot-learning.org/CoRL/2022/Conference — CoRL 2022 Poster_

### Official Review · Reviewer_GwM5 · 2022-07-24

**Originality:** Very Good
**Technical Quality:** Very Good
**Clarity Of Presentation:** Very Good
**Impact:** 4

**Recommendation:**

Strong Accept: I recommend accepting the paper and will argue for my recommendation even if other reviewers hold a different opinion.

**Summary:**

This paper tackles an interesting problem on enabling robot teachers to teach human students in a multi-agent cooperative setting through interaction. The authors formalize the problem, and propose a general conceptual framework along with one simplified instantiation with it. The method is evaluated in two tasks, overcooked in simulation and duo maze ball with robots

**Issues:**

- Typo: figure 2 caption: ‘represnet’ should be ‘represent’

For further issues, please refer to the weakness.


**Quality Of The Limitations Section:**

Limitations are addressed clearly

**Reviewer Expertise:**

4: The reviewer is confident but not absolutely certain that the evaluation is correct

**Robotics Focus:**

Sufficient demonstration on hardware

**Strengths And Weaknesses:**

Strengths:
- The paper is well written. The motivation is very clear and straightforward. Method description is very detailed. Although it has a lot of notations since it formalizes the problem, they are all very clear.
- The problem itself is novel and interesting. The paper focuses on a novel setting where the robot teaches the human partner to cooperate through interaction.
- Although the specific instantiation of the framework makes certain assumptions and simplifies the problem, it manages to model and reason over the adaptive human, which inherently is challenging itself.


Weaknesses:
- More reference is needed for motivation the problem. Even though the motivation is straightforward, adding more reference to support this might be better. Some examples are:
* In the introduction, “the learner can no longer learn to cooperate by simply observing the demonstration” what are the drawbacks of directly learning from multi-agent demonstrations?
* In the introduction, “human normally learns through interacting and practicing the skills with representative teachers, rather than exhaustively collecting demonstrations to cover all possible behaviors of the partners” Any literature to support this claim as well?
- Adding explicit statement of all the assumptions earlier might be clearer in the first place. E.g. the robot (teacher) assumes full knowledge about the task, sub-skills decomposition etc.
- It somehow feels like the framework described in Section 3 is too general while the instantiation in Section 4 simplifies too much by contrast. I am not against a general setup and do appreciate that this paper tries to formalize the problem and propose a general framework. Maybe adding some discussion on potential instantiation/more application of this framework in appendix would make it stronger.



**Summary Of Recommendation:**

Overall, this paper is well written, looks at a novel problem and makes a solid first attempt in solving this by proposing a general conceptual framework and a practical instantiation.  I recommend for acceptance.

---

> ### Author Response · Authors · 2022-08-22
> **Q4 Adding some discussion on potential instantiation/more application.**
>
> >Maybe adding some discussion on potential instantiation/more application of this framework in appendix would make it stronger.
>
>
> Thanks for the great suggestion! We have added the following paragraph on potential instantiation/ more applications in Section E in the appendix.
>
> *We provide a conceptual framework for cooperative robot teaching and a practical solution to it. Teaching humans to learn to cooperate has a broad field of applications and each of them may require a dedicated solution based on the application. In the following, we provide some potential applications and possible solutions:*
>
> * *Training humans in cooperative sports. For example, the framework can be applied to train humans in learning table tennis doubles. The framework can be further extended to train humans in team games with robots as the team members.*
> * *The framework may also be applied to single-player tasks with proper division of the task. For example, Assisted Chinese calligraphy writing. We show one possible application of Chinese calligraphy writing in figure. In this task, the robot would write the Chinese characters together with humans. As there are specified strokes in the Chinese writing system, they provide natural sub-skill decomposition. Based on a human’s performance, the robot could decide on what is the next character to train the human on. In addition, a finer-grained curriculum can be devised to train humans. For example, varying robot’s assistance in terms of force exerted on the human hand to help the human write.*

---

> ### Author Response · Authors · 2022-08-22
> **Q3 Adding explicit statement of all the assumptions earlier might be clearer in the first place.**
>
> >Adding explicit statement of all the assumptions earlier might be clearer in the first place. E.g. the robot (teacher) assumes full knowledge about the task, sub-skills decomposition etc.
>
> Thank you and we totally agree. We have made the modification to make the assumptions in both the general formulation and the solution more explicit. We refer the reviewer to line 61 - line 64 and approach lines 186-189 in the revised manuscript.

---

> ### Author Response · Authors · 2022-08-22
> **Q2 Any literature to support this claim as well?**
>
> >Q2 “human normally learns through interacting and practicing the skills with representative teachers, rather than exhaustively collecting demonstrations to cover all possible behaviors of the partners”
>
> We thank the reviewer for pointing out this insight. We have included a reference from cooperative learning to support this statement in line 23 in the revised manuscript.

---

> ### Author Response · Authors · 2022-08-22
> **Q1 what are the drawbacks of directly learning from multi-agent demonstrations?**
>
> >In the introduction, “the learner can no longer learn to cooperate by simply observing the demonstration” what are the drawbacks of directly learning from multi-agent demonstrations?
>
> We thank the reviewer for bringing up this issue.
>
> *A multi-agent game is combinatorially more complex than a single-agent game. Collecting the demonstrations for a multi-agent game is much more difficult as we need interactive data among several agents, and finding a suitable model to simulate at least two agents itself is difficult [1]. In addition, to each agent, the environment is non-stationary with distinct optimal demonstrations[2]. The entire state/action space is combinatorially larger than a single-agent game.  The success of an LfD policy relies on the coverage and the quality of demonstrations. The difficulty in collecting interactive demonstrations, together with the enormously large space of all possible demonstrations, makes the direct imitation learning from demonstrations extremely difficult[3].*
>
> *[1] Hernandez-Leal et al., A Survey of Learning in Multiagent Environments: Dealing with Non-Stationarity, Arxiv, 2017.
> [2] Lowe et al, Multi-Agent Actor-Critic for Mixed Cooperative-Competitive Environments, NIPS 2017.
> [3] Song et al., Multi-Agent Generative Adversarial Imitation Learning, NeurIPS 2019.*
>
> We also added the discussion in the appendix in Section D.

---

> ### Author Response · Authors · 2022-08-22
> **Response to Reviewer GwM5**
>
> We thank the reviewer for the insightful reviews. The reviewer's suggestion is to take in more references to support the motivation. We detail the response in the following threads. Unless otherwise stated, line numbers correspond to those in the revised manuscript. We kindly ask the reviewer to let us know if further clarification or information is needed from us.

---

### Official Review · Reviewer_thAt · 2022-07-31

**Originality:** Very Good
**Technical Quality:** Good
**Clarity Of Presentation:** Excellent
**Impact:** 3

**Recommendation:**

Weak Accept: I recommend accepting the paper, but will not argue for my recommendation if the majority of other reviewers have a different opinion.

**Summary:**

This work first proposed a conceptual framework for the cooperative robot teaching where a robot(autonomous agent) is the teacher, and a human user is the student. The proposed framework allows the robot teacher to adaptively select actions in order to accelerate the student's learning process. The authors then implemented a version of the framework and evaluated the system in two simple games with 20 and 16 human participants respectively. In their implementation, a simple one-parameter model and the Wiener process was used to model the learner's state and the knowledge acquisition process. Maximum a Posteriori Estimation was used to control the robot by learning three critical parameters (i.e. task difficulty, student proficiency, and the learning "smoothness") from experience. The user study provided some evident showing the proposed method works better than a fully assistive and a random teacher baseline. Yet, the experiments only provided very limited amount of conclusive results.

**Issues:**

Questions and concerns are described in the weaknesses section.

**Quality Of The Limitations Section:**

Limitations are addressed clearly

**Reviewer Expertise:**

5: The reviewer is absolutely certain that the evaluation is correct and very familiar with the relevant literature

**Robotics Focus:**

Sufficient demonstration on hardware

**Strengths And Weaknesses:**

Strengths: This work presented an abstract framework for cooperative robot teaching, which specifically model the human learning process and allows the robot to select teaching strategy adaptively. The idea of robot teaching/coaching has been explored by a few existing HRI works, but has not yet been extensively studied. Therefore, the proposed framework provides an interesting research direction for the future works. The authors also provided a functional implementation of the frame, which can be used as a baseline system for the future works. The paper is well written, the ideas and concepts are clearly explained, making it very easy to follow.

Weaknesses: The proposed framework was tested in two simple environments with very small action, state, and parameter space. Applying this framework to a more difficult task can be very challenging, which is the major limitation of this work. Because accurately initializing the parameters and estimating the hidden human state can be drastically more difficult and data demanding as the task complexity increases. Moreover, the reviewer has questions and major concerns over the experiments:

What is the definition of "efficient strategy" in the two games and how was it identified during the experiments?

It is not clear to the reviewer what exactly do the figure 4b and the discuss on the percentage of reward(line 278 ~ 280) are trying to show and how do they support the argument about the fully-assistive teacher impedes learning.

In figure 4.a., any insights about why the students perform significantly worse in evaluation than in training in the fully-assistive scenario? With increased experience, one would expect a similar or better performance in simple task learning.

The claim in line 285, "random teacher is incapable of teaching a truly cooperative task", is not supported by the experiment. As the author stated, the Duo Ball Maze task requires at least some degree of cooperation between the two players to successfully finish the task. This means that the random agent is more of a disturbance generator instead of a cooperative helper. Drawing conclusions about if the human users have learned COOPERATIVE skills by directly comparing these two settings is not sufficient. Instead, a better experiment could be training the users via the two different teachers, and evaluating the users' skills with another unseen baseline cooperative robot.

Similarly, the statement in line 298, "Such a strategy…, therefore resulting in a poorer performance score", suffers from the same problem where the human performance of different settings are not directly comparable when evaluated by different robot partners.

For figure 6, how were the line plots created? And can you explain why are these individuals representative enough to speak for the whole population? Could this figure be replaced by some statistics that summarize the entire group?


**Summary Of Recommendation:**

This paper presented a framework that allows a robot to assist human skills learning via cooperative HRI. The proposed idea is interesting, the system was well implemented, and the experiments have sufficient amount of human samples. Yet, the experimental results provided in the paper do not fully support some of the important claims. The conclusive results that can be draw from the current experiments are relatively weak.

I appreciate the authors' detailed responses. The biggest concern of mine about insufficient experimental setup appeared to be a misunderstanding on the experimental procedure. I am changing my recommendation to weak accept.

---

> ### Author Response · Authors · 2022-08-22
> **Q6 For figure 6, how were the line plots created?**
>
> >Q6: For figure 6, how were the line plots created? And can you explain why are these individuals representative enough to speak for the whole population? Could this figure be replaced by some statistics that summarize the entire group?
>
>
> We created the line plots by smoothing the raw performances (the dots) over interactions. We thank the reviewer for raising this question and have modified the caption of Figure 6.
>
> We do not intend to represent the whole population using just two participants. We want to qualitatively highlight that our proposed formulation can design an individualized curriculum. Participants 4 and 6 are the extreme cases: the student improves the most when trained in a particular sub-skill, and the teacher allocated the most training iterations to that sub-skill. In the appendix, we present all participants' raw and smoothed performances in Figures 2-4 (lines 114-118).

---

> ### Author Response · Authors · 2022-08-22
> **Q5 In figure 4.a., any insights about why the students perform significantly worse in evaluation than in training in the fully-assistive scenario?**
>
> >Q5:In figure 4.a., any insights about why the students perform significantly worse in evaluation than in training in the fully-assistive scenario? With increased experience, one would expect a similar or better performance in simple task learning.
>
> During the evaluation, the students in all three groups are evaluated with the same previously unseen robot partner. The behavior of this robot partner is not perfectly assistive, thus, leading to the degradation of the performance of the student in the evaluation round.

---

> ### Author Response · Authors · 2022-08-22
> **Q4 It is not clear to the reviewer what exactly do the figure 4b**
>
> >Q4:It is not clear to the reviewer what exactly do the figure 4b and the discuss on the percentage of reward(line 278 ~ 280) are trying to show and how do they support the argument about the fully-assistive teacher impedes learning.
>
> That is a great question for us to clarify! In Figure 4(a), we show that the students with a student-aware teacher outperform those with a fully-assistive teacher in the evaluation stage. In fact, students of the fully-assistive teacher perform the worst during evaluation.
>
> One may hypothesize that the student becomes too “lazy” to **act** because the fully-assistive teacher would do everything, hence the student has fewer opportunities to practice. However, the result shown in Figure 4(c) conflicts with that hypothesis. Figure 4(c) shows the percentage of reward contributed by the student during training; the students of a fully-assistive teacher actually contribute a high percentage of the reward. Figure 4(b) shows the percentage of students who found the efficient strategy, and few students of the fully-assistive teach found that strategy, indicating that the students didn’t explore. Figure 4(b)-(c) help us understand why a fully-assistive teacher could impede learning: The fully-assistive teacher may disincentivize the students from **exploring new strategies** rather than **executing actions**.

---

> ### Author Response · Authors · 2022-08-22
> **Q3 What is the definition of "efficient strategy"**
>
> > Q3:What is the definition of "efficient strategy" in the two games and how was it identified during the experiments?
>
>
> Thank you for pointing this out. The efficient strategy only exists for the Overcooked experiment and it is determined by the layout of the game. We briefly introduce such an efficient strategy in line 260 of the original manuscript and we realize that it is not clear enough. In short, the efficient strategy is 1) put multiple onions on the middle table; 2) go to the pot; 3) pick up onions from the middle table; 4) put them into the pot,  rather than picking up one onion at a time and put them on the pot. The overall idea is to reduce the number of movements needed to deliver the same amount of ingredients to the pot. We refer the reviewer to Section B.1 in the appendix for demonstrations. We also highlight the change in the modified version of the manuscript to make it more accessible to the readers in lines 277-280.

---

> ### Author Response · Authors · 2022-08-22
> **Q2 The proposed framework was tested in two simple environments with very small action, state, and parameter space.**
>
> >Q2: The proposed framework was tested in two simple environments with very small action, state, and parameter space. Applying this framework to a more difficult task can be very challenging.
>
> OverCooked and Duo Ball Maze are, in fact, not simple tasks to learn. The difficulty of teaching is jointly determined by the size of the state/action space, the transition function, and the reward function. The optimal plan for a task with small space but a time-varying dynamic function and sparse reward may be much more difficult than a task with enormous state/action space but simple dynamics and dense reward function. Though overcooked is of small state space, the optimal best-response policy varies significantly to different partners and the layout of the game is designed purposely to make the coordination challenging[1]. In addition, the Duo Ball Maze is of continuous state/action space of dimension 4 and 2 respectively. Such a space is large even for a control task in simulation.
>
> Further, we want to test our approach on simplified settings as a first step. This is common practice in HRI[2,3,4], in order to remove the possible confounders to the main hypotheses.  As a result, we can use the task success rate to directly evaluate the effectiveness of the teaching policy. We completely agree that applying the framework to a more realistic, complex task is the ultimate test of success. However, a more complex machine teaching task may involve complexities in perception, planning under uncertainty, bounded rationality, different reactions in risky situations, and control. These challenges exist even without the difficulty of transferring the knowledge from a machine teacher to human students with different capabilities. If we test our framework under such a complex task setting, we would not be able to distinguish the effectiveness of the teaching policy from the other possible aspects that would significantly affect the final student performance.
>
> [1] Carroll et al., On the utility of learning about humans for human-ai coordination, NeurIPS 2019.
> [2] Stefanos Nikolaidis and Julie Shah, Human-Robot Cross-Training: Computational Formulation, Modeling and Evaluation of a Human Team Training Strategy, HRI 2013.
> [3] Jarrasse et al., A Framework to Describe, Analyze and Generate Interactive Motor Behaviors, PLOS one, 2012.
> [4] Li et al., A Framework of Human-Robot Coordination Based on Game Theory and Policy Iteration, TRO, 2016.

---

> ### Author Response · Authors · 2022-08-22
> **Q1 The human performance of different settings are not directly comparable when evaluated by different robot partners.**
>
> >Q1:The human performance of different settings are not directly comparable when evaluated by different robot partners.
>
> Human performance is evaluated with the same previously unseen robot partner.  In both experiments, we “train the users via the two different teachers, and evaluate the users' skills with another unseen baseline cooperative robot”.  In the evaluation round, students from different groups are tested against a common previously unseen robot partner. We realized that we did not make the point clear and thank the reviewer for raising the issue. We provide a brief explanation here: For Overcooked, the unseen partner is generated by training a sub-optimal partner with population-based training; For Duo Ball Maze, the unseen partner has pre-specified compliances and the ball is randomly put on either side of the board, and we set compliances as the means of the uniform distributions from which the robot’s compliances are sampled during training.
>
> For details on how the evaluation is done, we refer the reviewer to Section B.1 B.2 in the appendix. We also have made modifications correspondingly to make this point explicit in line 282.
>
> **We now address the remaining questions.**

---

> ### Author Response · Authors · 2022-08-22
> **Response to Reviewer thAt**
>
> We thank the reviewer for the helpful reviews. The concerns of the reviewer are surrounding primarily (i) the generalization of the proposed framework to difficult tasks and (ii) interpretation of the experiment results. **We want to highlight a possible misunderstanding on the experiment setup and answer that question first.** Unless otherwise stated, line numbers correspond to those in the revised manuscript. We kindly ask the reviewer to let us know if further clarification or information is needed from us.

---

### Official Review · Reviewer_aCDa · 2022-08-02

**Originality:** Good
**Technical Quality:** Poor
**Clarity Of Presentation:** Poor
**Impact:** 2

**Recommendation:**

Strong Reject: I recommend rejecting the paper and will argue for my recommendation even if other reviewers hold a different opinion.

**Summary:**

This work proposes a framework for using robot experts to teach human users to optimize collective rewards in collaborative tasks. This type of machine teaching is novel, in that in this scenario, the robot is the agent with expert knowledge that is to be transferred to the human, who is assumed to be initially inexperienced in the given task.

The proposed framework combines elements of active teaching — making assumptions that the robot teacher has full knowledge of the human’s optimal policy and can select meaningful states to use to teach the human — as well as elements of hierarchy and item response theory to decompose the behavior of the human into a set of a priori defined “sub-skills” that the robot teacher can decide between to teach the student.

The results show that student-aware teachers that reason over sub-skills are slightly better or on par with “fully assistive” teachers, and are slightly better than “random” teachers.

**Issues:**

It seems that as a whole, the key component of this work that makes it better than random approaches or fully assistive approaches is the partitioning into sub-skills — not any of the added complexity of the item response theory or knowledge tracing components. The punchline of this paper seems like it can be crystallized as “in collaborative tasks with clear divide-and-conquer strategies, we can teach humans efficiently by teaching them to exploit one of the strategies, while the robot does the other.” This is not general to even normal collaborative HRI tasks (consider assistive feeding or collaborative construction), let alone broader tasks. The assumption that we can divide up a task into a small K <= 2 number of sub-skills is too strong, and coupled with the limited evaluation, and poorly justified algorithmic complexity, raise many issues I hope the authors can address in future submissions.

Minimally, I’d love to see a stronger set of baselines (with various heuristic “fully assistive” teachers), a set of ablations or simplification of the current algorithm for teaching, and more robust evaluation metrics.

**Quality Of The Limitations Section:**

Limitations are addressed clearly

**Reviewer Expertise:**

4: The reviewer is confident but not absolutely certain that the evaluation is correct

**Robotics Focus:**

Relevant but unlikely to deploy to hardware in near future

**Strengths And Weaknesses:**

I think the motivation of this work is nice — I strongly believe that as our agents become more capable, they will play a huge role in teaching humans new skills; I think developing general frameworks to that end is immensely useful.

---

Unfortunately, I have some serious concerns with this work, stemming from the results and the initial setup and assumptions of the framework.

Regarding the results — I find it hard to justify the differences between the “fully assistive” baseline and the “student aware” (proposed approach) in both Figures 4 and 5. When it comes to a question of reward, the student aware baseline is not ever significantly better than the fully assistive strategy, and the graphs that do show a significant improvement (e.g., Figure 4b) are estimated over metrics such as “percentage of students who found the efficient strategy” — where efficient is defined as a factor of the underlying hierarchy that is only exposed to the student-aware agent! Furthermore, the “fully assistive” baseline is rather weak (and already hugely performant) — most “assistive” baselines for these types of dynamic/adaptive tasks look at annealing the amount of assistance that the robot gives over time either with a fixed, or learned schedule... approaches that seem like they’d significantly beat the student-aware policy by a sizable amount, given these results (and those in the appendix). I strongly believe this work needs to compare against these simple heuristic baselines to justify the complexity of the proposed approach.

The approach itself also gives me cause for concern — there are several complex moving parts, without meaningful justification of why each component is necessary. To start with, Definition 2 around the teaching task makes a *huge assumption* that the robot teacher has full knowledge of the oracle human policy, and is working to guide the human to that policy as fast as possible... however, it feels strange to me that there’s only “one” optimal policy that does not take into account the preferences of the user at all — indeed, this seems to be a huge hurdle for collaborative leader/follower tasks like those studied in this work.

The second major concern I have with the approach is the formulation of the teaching task as having the robot pick from a set of pre-defined skills, via Item Response Theory; Section 4.2 states that the sub-skills K is defined a priori (for example, in the OverCooked Task the skills are “putting ingredients into the pot”, and “delivering the soup” and for the real robot task, following a similar 2 sub-skill “divide-and-conquer” strategy). Furthermore, the parameters for the Item Response Theory component that the teacher uses to figure out what skills to teach the robot are not well-defined; 4.2 states that the difficulty parameter Beta and the Smoothness parameters Lambda are fixed constants... but Algorithm 1 indicates that these are all learned in tandem? How does this happen — none of the details are here, and a naive optimization of these parameters could lead to arbitrary outcomes.


**Summary Of Recommendation:**

Based on the lack of motivation and underspecification of the core algorithmic components of this work, the limited evaluation with lukewarm results, and the huge assumptions made about the knowledge of the robot teacher and tasks, I would advocate that the paper be rejected at this time.

---

> ### Author Response · Authors · 2022-08-22
> **Q8 The assumption that we can divide up a task into a small K <= 2 number of sub-skills is too strong.**
>
> >Q8: The assumption that we can divide up a task into a small K <= 2 number of sub-skills is too strong.
>
> While in this work, we show experiments with K=2 sub-skills, the solution can be applied to more general cases with K>2. During each interaction, the robot teacher performs a subset of the K sub-skills and leaves the remaining ones to the student to practice. To illustrate, we consider a task that can be decomposed into K=4 sub-skills. The robot and the student can perform two of them respectively. In each round of interaction, the human is trained for two sub-skills, decided by the robot teacher. The robot teacher makes decisions based on the history of the student’s performances. Such a strategy is similar to presenting students with a question associated with multiple key concepts[1]; in our case, each round of training is associated with multiple sub-skills.
>
> [1] Lan et al., Sparse Factor Analysis for Learning and Content Analytics, Journal of Machine Learning Research 15, 2014

---

> > ### Comment · Reviewer_aCDa · 2022-08-26
> > **Q8 Response**
> >
> > I absolutely agree that you *have the ability* to scale to more skills with this framework, and this argument makes sense; I just would've loved to see this in the experiments. It would also help mitigate a lot of the problems I hinted at in Q7!

---

> ### Author Response · Authors · 2022-08-22
> **Q7  we can teach humans efficiently by teaching them to exploit one of the strategies, while the robot does the other.**
>
> > Q7: It seems that as a whole, the key component of this work that makes it better than random approaches or fully assistive approaches is the partitioning into sub-skills — not any of the added complexity of the item response theory or knowledge tracing components. The punchline of this paper seems like it can be crystallized as “in collaborative tasks with clear divide-and-conquer strategies, we can teach humans efficiently by teaching them to exploit one of the strategies, while the robot does the other.”
>
> We respectfully disagree. “in collaborative tasks with clear divide-and-conquer strategies, we can teach humans efficiently by teaching them to exploit one of the strategies, while the robot does the other” is the description of the “random teacher” baseline for our Duo Ball Maze experiments as illustrated in the original manuscript line 253, where the teacher would randomly choose a sub-skill to teach. We also conducted a post-experiment survey to understand human preferences in the original manuscript lines 312 to line 314. The result indicates that similar to other educational domains, teaching a cooperative skill benefits from an adaptive curriculum. In Figure 6, we show how the provided solution generates a personalized curriculum. We also show in Figure 5(c) that individualization facilitates human learning.
>
> However, we understand the presentation may cause confusion and thank the reviewer for pointing it out. We have revised the manuscript accordingly and refer the reviewer to lines 269-270.

---

> > ### Comment · Reviewer_aCDa · 2022-08-26
> > **Q7**
> >
> > Thanks for the clarification and update to the manuscript. I definitely think it reads clearer now.
> >
> > As for the "random teacher" baseline in Duo Ball Maze already matching the description of the problem; I agree! That's a bit why I'm so concerned with this point; for a random baseline that teaches a fixed skill, the evaluation performance is pretty high (and better than a fully-assistive strategy)! Isn't this strange? Especially given that the 95% CIs for the student-aware approach and the random approach overlapping? Does this not mean that it's perhaps optimal under the given set of assumptions to just resort to divide and conquer - except with some limited assistance?

---

> > > ### Author Response · Authors · 2022-08-27
> > > **Q7 follow-up**
> > >
> > > We think the result is reasonable (and the random teacher is not better than the fully-assistive strategy in Duo Ball Maze). As once the task is decomposed, the complexity of cooperative learning can be effectively reduced because students can focus on restricted sub-problems during training [1]. Even trained with a random teacher, the human student can quickly adapt and learn. The *divide-and-conquer* technique is the basis of efficient algorithms for many problems and is important for solving complex tasks. However, *divide-and-conquer* itself does not tell us 1) how we *divide* the problem and 2) how we *conquer* the sub-problem. In this work, we *divide* the skill with role allocation and *conquer* each of them within limited teaching time through online proficiency estimation and partial assistance.
> > >
> > > [1] Wang et al., RODE: Learning Roles to Decompose Multi-Agent Tasks, ICLR 2021.

---

> ### Author Response · Authors · 2022-08-22
> **Q6 however, it feels strange to me that there’s only “one” optimal policy that does not take into account the preferences of the user at all.**
>
> > Q6: however, it feels strange to me that there’s only “one” optimal policy that does not take into account the preferences of the user at all — indeed, this seems to be a huge hurdle for collaborative leader/follower tasks like those studied in this work.
>
> We thank the reviewer for pointing this out. We completely agree that a student may have multiple optimal policies $\Pi^* = \{\pi^*_1, \pi^*_2, …\}$, and the choice of which student policy to teach is an important question.  A good choice of the student policy may reduce the teaching effort and also improves the student's learning experience. A principled approach to selecting the optimal student policy needs to consider the student's preference, him/her update model for the knowledge level, and an estimate of his/her current capability, which requires insight beyond our work. In this paper, we assume that we have an oracle to choose the optimal student policy $\pi^*  \in \Pi^*$ to teach, such that this policy $\pi^*$ matches the preference of the student and also reduces the teaching effort.

---

> > ### Comment · Reviewer_aCDa · 2022-08-26
> > **Response Q6**
> >
> > This is the biggest assumption of the paper - one that I don't agree with; I don't know that for the tasks evaluated on (and for the motivating tasks in general in robotics) that you can assume a single oracle policy to teach. This is the crux of the arguments Q2-Q4 above.

---

> > > ### Author Response · Authors · 2022-08-27
> > > **Follow-up on Q6**
> > >
> > > We have revised the manuscript (see revised manuscript line 137) to account for multiple optimal policies and we do not assume a single policy. It is only during teaching, the teacher will select a single student policy to teach based on the student’s preference. If we want to teach multiple optimal student policies, we can teach them one at a time. In fact, teaching a single student policy at a time is more realistic than learning all optimal policies at once. Firstly, unlike machine agents, a human agent prefers to master a single optimal policy, rather than knowing every way to solve the tasks. In addition, even if we wish to teach multiple tasks, teaching one at a time is more intuitive for the human student: the multiple optimal policies may have different action choices at the same state, teaching them all at once will likely confuse the student for this multi-modality.

---

> ### Author Response · Authors · 2022-08-22
> **Q5 annealing the amount of assistance seem like they’d significantly beat the student-aware policy by a sizable amount.**
>
> > Q5 most “assistive” baselines for these types of dynamic/adaptive tasks look at annealing the amount of assistance that the robot gives over time either with a fixed or learned schedule... approaches that seem like they’d significantly beat the student-aware policy by a sizable amount.
>
> Yes, we agree that a teacher “annealing the amount of assistance” could be a stronger baseline than the fully-assistive one based on our findings. However, when “annealing the amount of assistance”, the teacher becomes partially assistive instead of fully assistive. We need a quantitative measure of assistance. This measure is task-dependent and difficult to quantify in general. Taking the Overcooked experiments as an example: How is the amount of assistance measured? What is the granularity of experiments required to tell the difference between different partially assistive policies?

---

> > ### Comment · Reviewer_aCDa · 2022-08-26
> > **Q5 Response**
> >
> > I'm confused; in the updated manuscript on Line 265, the "fully-assistive" teacher "... performs optimally concerning the student's *initial* capability" - can't we update the fully-assistive teacher to either update it's belief over the student's initial capability over time, and start out more random (similar to an epsilon-greedy strategy?). This would allow the student to express themselves/learn well against the random policy initially before figuring out where they are being suboptimal.
> >
> > Your quantitative measure would be the number of "teacher updates" over the course of training; I'm less concerned with having an exhaustive/granular set of experiments but rather just 1-2 "sanity check" experiments to motivate the need for a more complex partially assistive algorithm; for OverCooked, running out one update in the middle of the 5 games trained on (or 2 updates across 6 games) would make sense; for Duo Ball Maze, you have 20 interactions -- re-evaluating in the middle, or after every 4 interactions could be meaningful?

---

> > > ### Author Response · Authors · 2022-08-27
> > > **Follow-up on Q5**
> > >
> > > Thank you for acknowledging that a teacher “annealing the amount of assistance” is partially instead of fully assistive. We believe that a teacher “annealing the amount of assistance” is indeed interesting and intuitive; we will definitely have it in our future work. Still, we would like to emphasize that we base our approximate solution on the skill-decomposition with an intention to reduce the policy space so as to speed up the convergence of the student. “Annealing the amount of assistance” optimizes over the entire policy space, while the skill-decomposition naturally focuses on a few small subspaces of the entire policy space. As a result, for the teacher who “anneals the amount of assistance”, the difficulty of teaching is proportional to the size of the policy class; while for us, we have the ability to moderate the difficulty via refining the policy space by skill decomposition. Therefore, when the complexity of the task scale up, “annealing the amount of assistive” will likely to suffer from the “curse of dimensionality”, but not us.
> > >
> > > Decompositional approach to teaching is also well established and recognized in both educational research[1] and real-world practice. For example, in the driving school, the student learns sub-skills like S-curve driving, reverse parking, parallel parking, etc.
> > >
> > > [1]Grossman et al., Teaching Practice: A Cross-Professional Perspective, Teachers College Record: The Voice of Scholarship in Education,2009

---

> ### Author Response · Authors · 2022-08-22
> **Q4 the graphs that do show a significant improvement (e.g., Figure 4b) are estimated over metrics such as “percentage of students who found the efficient strategy”**
>
> >Q4 the graphs that do show a significant improvement (e.g., Figure 4b) are estimated over metrics such as “percentage of students who found the efficient strategy” — where efficient is defined as a factor of the underlying hierarchy that is only exposed to the student-aware agent!
>
> Thank you for your questions. Can you please clarify what “...where efficient is defined as a factor of the underlying hierarchy …” mean?
>
> Our “efficient strategy” is defined over a pre-specified desired behavior. For example, in OverCooked, the “efficient strategy” is the following sequence of actions: 1) put multiple onions on the middle table; 2) go to the pot; 3) pick up onions from the middle table; 4) put them into the pot. This efficient strategy only depends on the task; it stays invariant to the change in role decomposition. We refer the reviewer to Section B.1 and Figure 1 in the supplementary materials for demonstrations. This “efficient strategy” is NOT provided to the students from any teaching group. Hence, our student-aware teaching strategy does not benefit unfairly from efficient demonstrations, nor do we use it as the main metrics to compare students’ performance.

---

> > ### Comment · Reviewer_aCDa · 2022-08-26
> > **Q4 Response**
> >
> > So when I say "efficient is defined as a factor of the underlying hierarchy" I think I mean exactly what the response says -- that "efficiency" is measured relative to a pre-specified behavior (one that is integrated into the skills defined for the IRT, to the class of optimal policies that the teacher has knowledge over, to the teacher reward function).
> >
> > I know that this hierarchy/behavior is never specified to the students; I think the theme of Q2/Q3 and this question is that I'm very worried that injecting this "pre-defined hierarchy" into all the other parts of the proposed teaching system collapses the space of possible student policies that could get high reward.
> >
> > I do believe that for a cooperative robot teaching policy to succeed, the goal is less about top-down transfer from teacher to student that **stifles the student policy**, but rather adapting to and correcting for human suboptimality; this is what all my questions and response so far have been trying to explicitly get at.

---

> > > ### Author Response · Authors · 2022-08-27
> > > **Follow-up on Q4**
> > >
> > > Thank you for acknowledging our clarification that the efficient strategy is exposed to students from all groups. After skill-decomposition, only a subset of the policy space can be covered by the decomposed skills. We understand the reviewer’s concern is regarding a bad skill-decomposition leading to the collapse of space of possible student policies. Granted that the optimal policies may lie outside the space covered by the decomposed skills, we would like to point out that this is the issue of the bad choice of skill-decomposition, instead of the skill-decomposition itself. In fact, we can show that there always exists a skill decomposition covering the optimal policies. For example, in the extreme case, the skill decomposition only contains a single skill, which is the single optimal policy. We discuss the limitation of manually-defined skill-decomposition in the Limitation section.

---

> ### Author Response · Authors · 2022-08-22
> **Q3 The second major concern I have with the approach is the formulation of the teaching task as having the robot pick from a set of pre-defined skills, via Item Response Theory.**
>
> > The second major concern I have with the approach is the formulation of the teaching task as having the robot pick from a set of pre-defined skills, via Item Response Theory.
>
> Sub-skill decomposition is commonly practiced in pedagogical research and also the design for the commercialized online tutoring system[1] or even for general complex tasks. This decomposition of key concepts/sub-skills has been proven more effective than teaching the skill/knowledge directly, especially for complex skills[2, 3].
>
> In this work, we focus on the overall teaching framework. The sub-skills are manually defined in a task-dependent manner: they are therefore interpretable and easily visualized. In future work, we could consider automatic skill discovery[4] and role allocation[5], which allows us to decompose more complex tasks, e.g., multi-agent collaboration.
>
> >Furthermore, the parameters for the Item Response Theory component that the teacher uses to figure out what skills to teach the robot are not well-defined. By 4.2 states that the difficulty parameter Beta and the Smoothness parameters Lambda are fixed constants... but Algorithm 1 indicates that these are all learned in tandem?
>
>
> Thanks for pointing this out. We consider $\beta$ and $\lambda$ to be fixed but unknown to the teacher. A better description would be “both $\beta$ and $\lambda$ are learned without considering their change over time” and we have made the corresponding changes in the revised manuscript. For the optimization, we follow existing literature on the online estimation of student proficiency[6][7] in the Intelligent Tutoring System. We refer the reviewers to the reference and Section C in the appendix for a detailed description of how the optimization is done.
>
>
> [1] Knewton Inc., Knewton adaptive learning, 2015.
> [2] Corbett, A. T. and Anderson, J. R., Knowledge Decomposition and Subgoal Reification in the ACT Programming Tutor, Artificial Intelligence and Education, 1995.
> [3] Joseph E. Beck and Jack Mostow, How Who Should Practice: Using Learning Decomposition to Evaluate the Efficacy of Different Types of Practice for Different Types of Students, Intelligent Tutoring Systems, 2008.
> [4] Krishnan et al, DDCO: Discovery of Deep Continuous Options for Robot Learning from Demonstrations, CoRL, 2017.
> [5] Wang et al., RODE: Learning Roles to Decompose Multi-Agent Tasks, ICLR 2021.
> [6] Chaitanya Ekanadham, Yan Karklin, T-SKIRT: Online Estimation of Student Proficiency in an Adaptive Learning System, Machine Learning for Education Workshop at ICML 2015.
> [7] Wilson et al., Back to the basics: Bayesian extensions of IRT outperform neural networks for proficiency estimation, EDM, 2016
>
> **We now address the remaining questions.**

---

> > ### Comment · Reviewer_aCDa · 2022-08-26
> > **Q3 Response**
> >
> > Thank you for the clarifications and the revision in the paper - the role of beta and lambda are much clearer! I'm still not sure if pre-selecting from a known library of skills limits the space of good/optimal policies that students can learn though, but I this response makes sense to me.

---

> > > ### Author Response · Authors · 2022-08-27
> > > **Follow-up on Q3**
> > >
> > > Thank you for acknowledging our clarifications! We understand the reviewer’s concern is regarding a bad skill-decomposition leading to the collapse of space of possible student policies. Granted that the optimal policies may lie outside the space covered by the decomposed skills, we would like to point out that this is the issue of the bad choice of skill-decomposition, instead of the skill-decomposition itself.

---

> ### Author Response · Authors · 2022-08-22
> **Q2 Definition 2 around the teaching task makes a huge assumption that the robot teacher has full knowledge of the oracle human policy,**
>
> >Q2 Definition 2 around the teaching task makes a huge assumption that the robot teacher has full knowledge of the oracle human policy, and is working to guide the human to that policy as fast as possible...
>
> It seems that the reviewer is suggesting a “universal teacher” without specific knowledge of the targeted teaching outcome, in this case, the best policy. With all the evidence in the education research and in real-world pedagogical practice, such a universal teacher is most likely to be ineffective. Having a knowledgeable teacher is also a fair and necessary assumption for the work in the field of machine teaching[1,2,3]. In fact, a machine teacher who does not have this capability will be an odd setting: if the teacher does not know the optimal outcome, how can we trust he/she to teach humans? Similar to the standard setting of imitation learning, where it is well justified to have optimal expert human demonstrations for the machine to learn, a machine teacher needs to have the full knowledge of the task in order to teach humans.
>
> Our general problem formulation for robot teaching is based on this assumption of a knowledgeable teacher; however, we relax this assumption in our algorithmic implementation. Our problem definition aims to have a general and flexible formulation for the teaching task; therefore, we rely on the assumption of a knowledgeable teacher who knows the optimal policy. The optimal policy representation supports the encoding of a wide variety of teaching signals, be it optimal demonstrations, instructions, or reward functions. However, when we use our problem formulation to derive a solution, we relax this assumption in our algorithmic implementation by instantiating the optimal policy with a few optimal performances. The reviewer can refer to Section 4 for how we relax the assumption and section 4.4 on how to use the distance between the performances to approximate the distance between the policies.
>
> [1] Xiaojin Zhu, Machine teaching: An inverse problem to machine learning and an approach toward optimal education, AAAI 2015.
> [2] Liu et al., Iterative Machine Teaching, ICML 2017.
> [3] Liu et al., Towards Black-box Iterative Machine Teaching, ICML 2018.

---

> > ### Comment · Reviewer_aCDa · 2022-08-26
> > **Q2 Response**
> >
> > This is a really helpful clarification - thanks! I'm not suggesting a "universal teacher" but rather, realizing that knowing a single "optimal policy" (or having a single teaching task reward function) and guiding students towards that behavior is not necessarily the right thing to do in the case of environments with asymmetries or multimodal policy classes.
> >
> > I understand that the knowledgeable teacher assumption is used throughout the machine teaching literature, but these are all parameterized in more strict tasks where there's a clear (and singular) right answer. In the OverCooked and Duo Ball Maze environments, that this assumption holds up doesn't seem to ring true.

---

> > > ### Author Response · Authors · 2022-08-27
> > > **Follow-up on Q2**
> > >
> > > Thank you for recognizing that the assumption of “a knowledgeable teacher” is common in machine teaching literature! In fact, “a knowledgeable teacher” is similar to a “knowledgeable expert”, in the sense that both are assumed to know the optimal policies. The “knowledgeable expert” is common to the general machine learning literature; this assumption goes way beyond the tasks that “are all parameterized in more strict tasks where there's a clear (and singular) right answer”. In fact, there are many tasks with ambiguous task success definition that requires a fully knowledgeable teacher [1,2,3,4].
> > >
> > > Thank you for raising an additional question on the limitation of using a single optimal policy. We would like to point to our revised manuscript line 137, where we assume there is a set of optimal policies. It is during teaching that the teacher will select an optimal student policy that best suits the student's preference. This is a more realistic teaching scenario, where a human student would not want to learn ALL optimal policies, but a single policy that he/she prefers. We definitely agree automatically selecting one optimal policy for teaching is an interesting and important issue for future investigation. Moreover, if we want to teach all optimal policies, we can teach them one by one.
> > >
> > >
> > > [1] Ross et al., A Reduction of Imitation Learning and Structured Prediction to No-Regret Online Learning, ICML, 2011.
> > > [2] Hadfield-Menell et al.,  Cooperative Inverse Reinforcement Learning, NIPS, 2016.
> > > [3] Kelly et al., HG-DAgger: Interactive Imitation Learning with Human Experts, ICRA 2019.
> > > [4] Spencer et al., Learning from Interventions: Human-robot interaction as both explicit and implicit feedback, RSS, 2020.

---

> ### Author Response · Authors · 2022-08-22
> **Q1 The student-aware baseline is not ever significantly better than the fully assistive strategy.**
>
> > Q1 When it comes to a question of reward, the student-aware baseline is not ever significantly better than the fully assistive strategy.
>
> We respectfully disagree. Student-aware strategies show 55.7% improvement in Overcooked and 21.7% improvement in Duo Ball Maze compared to the fully-assistive baselines. These improvements are significant. In addition, a student-aware strategy almost doubles the performance improvement of an untrained student in Duo Ball Maze. This shows that our student-aware teaching strategy is effective. While there is still a gap to reach expert performance, our primary objective here is to compare different teaching strategies and not to reach expert-level performance.

---

> > ### Comment · Reviewer_aCDa · 2022-08-26
> > **Q1 Response**
> >
> > Looking at just the means reported in Figures 4/5, this is true! But I'm a bit more worried about the 95% CI reported, and how both these results measure up relative to the "random" baseline; it's not clear to me that the Evaluation performance on Duo Ball Maze is statistically significant, and the performance of the Random agent vis-a-vis the student-aware agent is similarly troubling.

---

> > > ### Author Response · Authors · 2022-08-27
> > > **Follow-up on Q1**
> > >
> > > Thank you for the question! We would like to point out that, considering the difference of the students under the student-aware teacher compared to other teachers, the improvement compared to the fully-assistive teacher is marginally statistically significant with one-sided t-test p-value at 0.053;  the improvement compared to the random teacher is also statistically significant with one-sided t-test p-value at 0.016.

---

> ### Author Response · Authors · 2022-08-22
> **Response to Reviewer aCDa**
>
> We thank the reviewer for comprehensive comments. The reviewer's concerns are mainly surrounding the experiment result, the assumptions we have made, and the justification of the instantiated solution. We detail the response in the following threads. We will answer the three more important questions first. Unless otherwise stated, line numbers correspond to those in the revised manuscript. We kindly ask the reviewer to let us know if further clarification or information is needed from us.

---

> > ### Comment · Reviewer_aCDa · 2022-08-26
> > **Overall Rebuttal Response**
> >
> > I really appreciate the well-written, and well-structured rebuttal points from the authors. I have in-lined my responses, and I am still struggling with some of the key assumptions of this work (even in light of the added clarifications in the text). I am leaning towards keeping my current score - but I also know that the other Reviewers have differing takes on the paper!
> >
> > At this point, I'd love to engage in discussion with the other reviewers about the merits of this paper, and if the authors have any final clarifications/arguments to make about my responses, I'd love to hear them (I do realize the discussion period is almost over though, so I don't expect any new experiments/analyses - just arguments as to why my current position might be ill-founded).

---

### Author Response · Authors · 2022-08-22
**General response and the revised manuscript**

**Comment:**

We thank reviewers for the encouragement and insightful feedback. The reviewers found:
1. the problem “novel” (R-aCDa, R-thAt, R-GwM5) and “interesting” (R-GwM5),
1. the motivation “nice” (R-aCDa) and “clear” (R-GwM5), and
1. the work providing “an interesting research direction for the future works” (R-thAt) with “interesting insights” (AC).

The main changes to the manuscripts are summarized below, based on suggestions from the reviewers and the area chair.
1. Stated the assumptions made in the framework in both Section 1 and Section 4. (R-aCDa, R-thAt, R-GwM5)
1. Added reference to support the motivation in Section 1. (R-GwM5)
1. Amended the definition of the teaching task to account for multiple optimal policies in Section 3. (R-aCDa)
1. Provided clearer explanations of the experiment setup and interpretation of the result in Section 5. (R-aCDa, R-thAt)
1. Discussed more limitations of the proposed framework and solution with possible ways of mitigating them in Section 6 and reduced the amount of discussion on performance comparison. (R-aCDa, AC)

The revised manuscript and appendix are attached. We kindly ask the reviewers to let us know if further clarification or information is needed.


**Zip File:**

/attachment/bc55b86185385781394d3a41ec9a35e4b78acdeb.zip

---

### Author Response · Authors · 2022-08-28
**Summary of the revised manuscript, response and discussion**

We thank all the reviewers and the Area Chair for reviewing and recognizing the strengths of this work! As the author-reviewer discussion period is going to end soon, we would like to summarize the updates on the manuscript, our responses, and the discussion.
&nbsp;
1. **Assumptions**. We state the assumption for the framework and method in both Section 1 and Section 4 explicitly in the revised manuscript (line 61 - 64, line 186 - 188). Specifically, we highlight why “the fully knowledgeable teacher” is a reasonable assumption: it is common practice in pedagogy and important for efficient teaching. We also show how such an assumption can be relaxed for the tasks in our experiments in Section 4.4 (e.g. Knowing the optimal performance).  We clarified decompositional approach to teaching is also well established and recognized in educational research and real-world practice.
&nbsp;
1. **Evaluation**.
   * We clarified that the students are evaluated with a common unseen robot partner (line 282). One reviewer (thAt) may have a potential misunderstanding here.
   * Both experiments are challenging and not “simple”. Though Overcooked is of small state space, the game is designed purposely to make the coordination challenging. In addition, the Duo Ball Maze is of continuous state/action space of dimension 4 and 2 respectively. Such a space is large even for a control task in simulation.
   * We clarified that the improvement over the baseline result is significant. Student-aware strategies show 55.7% improvement in Overcooked and 21.7% in Duo Ball Maze compared to the fully-assistive baselines. In response to reviewer (aCDa), in Duo Ball Maze, we provided the p-value compared with baselines (p=0.053 and p=0.017).
&nbsp;
1. **Limitations**. We have expanded the section on limitations and added more discussions about the assumptions in the general teaching framework and choices of our approximate solution. We have also discussed how to alleviate these limitations in the future.

---

### Meta-Review · Area_Chair_YP6r · 2022-08-14

**Recommendation:** Accept (Poster)
**Confidence:** 3

**Metareview:**

Post discussion review
----------------------

The rebuttal does a good job of taking the reviewer comments seriously and using them to improve the paper.  The authors even went so far as to upload a revised manuscript with their changes. Additionally, they also engaged in a series of detailed discussions with the most critical reviewer.

The additions and edits in the new manuscript really help clarify some of the main assumptions and highlight the value of the work. I believe that while there may be valid discussions to be had about certain limitations or assumptions, this work still has merit to the field.



Original review
---------------


This paper presents a framework for machine teaching of cooperative tasks. It includes evaluations in simulation (Overcooked) and on a physical robot (Duo Ball Maze).


Strengths
* Reviewers agreed that this area of machine teaching is underexplored and that the work addresses an important and valuable task.

* Reviewers noted that the paper is clear and well explained.

* There are some interesting insights in this work, including that people with fully assistive robots are disincentivized to explore ("lazy student effect"), that people will learn from a random robot in tasks that are semi-independent but not in fully cooperative tasks.


Weaknesses
* Reviewers were concerned about significant assumptions that are required for the model to work. For example, the framework relies on the robot knowing the correct human policy and having full knowledge of the task. These seem like they would quickly be violated in real world settings. It would help to understand how the given approach would transfer to more realistic and complex situations, or at least in what settings these assumptions could hold.

* Reviewers critiqued the evaluations, noting that the tasks were simple and the results were only slightly better for the proposed framework. Reviewers suggested alternative baselines for both simulated and real world tasks. They also posed a number of clarification questions about how the results are presented.

* In general, the paper might benefit from being clearer about the limitations of this approach and how they might be mitigated in the future, rather than trying to unequivocally outperform baselines with the current implementation.

---

> ### Author Response · Authors · 2022-08-22
> **Response to the Area Chair**
>
> >Q1: Reviewers were concerned about significant assumptions that are required for the model to work. For example, the framework relies on the robot knowing the correct human policy and having full knowledge of the task. These seem like they would quickly be violated in real-world settings. It would help to understand how the given approach would transfer to more realistic and complex situations, or at least in what settings these assumptions could hold.
>
> We have clarified why “the fully knowledgable teacher” is a reasonable assumption: it is common practice in pedagogy. We have relaxed this assumption for real-world tasks in our experiments (Q2 from R-aCDa). We have justified the choice of skill-decomposition (Q3,Q8 from R-aCDa). The manuscript is revised to state more explicitly (Q3 from R-GwM5) the assumptions for the formulation (line 61 - line 64) and approach (line 186 - line 188).
>
> >Q2: Reviewers critiqued the evaluations, noting that the tasks were simple and the results were only slightly better for the proposed framework. Reviewers suggested alternative baselines for both simulated and real world tasks. They also posed a number of clarification questions about how the results are presented.
>
> We must say that OverCooked and Duo Ball Maze are not simple tasks to learn (Q2 from R-thAt). We show that the proposed approach outperforms the baselines by a wide margin: student-aware strategies show 55.7% improvement in Overcooked and 21.7% improvement in Duo Ball Maze compared to the fully-assistive baselines. These improvements are significant (Q1 from R-aCDa). Discussion with potential baselines is presented in our response (Q5 from R-aCDa). The experiment setup and result are clarified in our response (Q1 from R-aCDa; Q1,Q3,Q4,Q5,Q6 from R-thAt).
>
> >Q3:In general, the paper might benefit from being clearer about the limitations of this approach and how they might be mitigated in the future, rather than trying to unequivocally outperform baselines with the current implementation.
>
> Thank you for the suggestion. We have expanded the section on limitations and added more discussions about the assumptions in the general teaching framework and choices of our approximate solution. We have also discussed how to alleviate these limitations in the future.